# Metronidazole Modified-Release Therapy Using Two Different Polymeric Systems Gels or Films: Clinical Study for the Treatment of Periodontitis

**DOI:** 10.3390/pharmaceutics16091108

**Published:** 2024-08-23

**Authors:** Mônica Danielle Ribeiro Bastos, Tatiane Cristina Dotta, Beatriz Roque Kubata, Cássio do Nascimento, Ana Paula Macedo, Fellipe Augusto Tocchini de Figueiredo, Millena Mangueira Rocha, Maria Paula Garofo Peixoto, Maíra Peres Ferreira, Osvaldo de Freitas, Vinicius Pedrazzi

**Affiliations:** 1Department of Dental Materials and Prosthodontics Ribeirão Preto, Ribeirão Preto School of Dentistry, University of São Paulo, São Paulo 14040-904, Brazil; mo.bastos@hotmail.com (M.D.R.B.); beatriz.kubata@usp.br (B.R.K.); cassionasc@forp.usp.br (C.d.N.); anapaula@forp.usp.br (A.P.M.); fellipe.figueiredo@usp.br (F.A.T.d.F.); millenamrocha@usp.br (M.M.R.); pedrazzi@forp.usp.br (V.P.); 2Department of Pharmaceutical Sciences, Ribeirão Preto School of Pharmaceutical Sciences, University of São Paulo, São Paulo 14040-900, Brazil; peixoto.mariapaula@gmail.com (M.P.G.P.); maira@fcfrp.usp.br (M.P.F.); ofreitas@fcfrp.usp.br (O.d.F.)

**Keywords:** periodontics, metronidazole, pH

## Abstract

This study evaluated the efficacy of semisolid systems (gels) and films containing a combination of metronidazole (MTZ) and metronidazole benzoate after scaling and root-planing (SRP) for periodontitis. In total, 45 patients with stage I or II periodontitis were enrolled and divided into 3 groups: 1—SRP—control; 2—SRP + Film with MTZ; 3—SRP + Gel with MTZ. The pH of gingival crevicular fluid (GCF) before/after treatments, MTZ concentrations, and drug release using high-performance liquid chromatography were investigated. The effects were evaluated by longitudinal monitoring of clinical parameters (probing depth—PD, clinical attachment level—CAL, and bleeding on probing—BP). MTZ and MTZ-benzoate concentrations in the periodontal pocket and pH showed no statistical difference after application. SRP + Gel presented the lowest CAL values. For SRP + Film and SRP + Gel, higher PD values were observed at T0 compared to all groups. A relevant reduction in BP was observed in SRP + Film and SRP + Gel groups at all times compared to T0. Both therapies improved periodontal health compared to SRP alone, reducing PD and BP, and increasing CAL for the gel group, suggesting they are promising for periodontal disease treatment.

## 1. Introduction

Periodontal diseases are localized inflammatory reactions that affect the supporting structure of the teeth [1,2,3,4]. As the inflammation progresses and affects periodontal supporting tissues, a place known as a periodontal pocket is formed between the roots of the affected teeth and the soft tissue. In this protected niche, a subgingival biofilm is organized, and the inflammatory condition is called periodontitis, which can culminate with teeth loss [1,2,3,4,5].

Periodontitis conventional treatment consists of the mechanical removal of biofilm and supra/subgingival calculus through scaling and root planing (SRP) with hand or ultrasonic instruments [1,2,3,6,7]. However, severely committed periodontal sites, with very deep periodontal pockets that are of difficult access to manual instrumentation, might not present a positive response to conventional treatment [1,8]. In these cases, it is possible to make use of surgical techniques, or even antimicrobials. The administration of antimicrobial agents has been applied as an important complement of mechanical debridement, once the systemic or local use of antibiotics might eliminate or decrease specific periodontal pathogens [1,3,8,9].

In contrast with systemic therapy of antibiotics, the administration of antimicrobial agents directly on periodontal pocket might reduce some undesirable side effects [1,3,8]. When associated to a low delivery system, the local antibiotics allow the maintenance of effective concentrations of a therapeutic agent only on the site of application for long periods. In addition, the side effects risks and the development of bacterial resistance are decreased [1,3,4,10]. Fibers, films, microparticles, chip, strips, or semi-solid (gels) formulations prepared with biodegradable polymers have been the devices proposed for the administration of antimicrobial agents on periodontal therapies [8,11].

In the realm of treating periodontal disease, local release systems have proven to be highly effective, with films and semi-solid or injectable systems emerging as prominent methods [8,11]. Among these, films offer several advantages, as they can be tailored to specific pocket sizes and shapes, ensuring precise delivery to the targeted area. Moreover, their quick and easy insertion minimizes discomfort for patients. The material’s thickness and adhesiveness play a key role, allowing the films to remain submerged between the gum and tooth without disrupting the patient’s eating or hygiene habits [12,13,14,15,16].

Similarly, semi-solid (gel) delivery systems present a straightforward and minimally discomforting approach to administering treatment within the periodontal pocket. The fluid nature of the formulation enables it to spread effectively throughout the pocket’s interior. However, to prevent it from being washed out by crevicular fluid, the formulation must exhibit strong adhesiveness and/or undergo a phase change, transitioning into a more rigid or solid state within the pocket [4,9,11,17,18].

Among the antimicrobial agents proposed for the treatment of periodontitis, metronidazole has been shown to be adequate due to its action spectrum, restricted to strict anaerobic microorganisms, as well as having reduced side effects when compared with tetracyclines, for example, as selection of multiresistant bacteria or disturbance of the body’s normal microbiota [1,6,7].

As per the findings of two separate studies conducted by Löfmark et al., 2010 [19], and Soysa et al., 2021 [7], metronidazole has been established as the preferred treatment for anaerobic infections. Its efficacy against these specific pathogens, coupled with low rates of microbial resistance, favorable pharmacodynamics and pharmacokinetics, and minimal adverse effects, make it a cost-effective and reliable choice.

Regarding its mode of action, metronidazole functions as a potent chemotherapeutic agent, disrupting bacterial DNA synthesis and leading to cell death. When administered orally, it exhibits excellent absorption properties, enabling it to reach comparable concentrations in plasma, saliva, and gingival fluid [17]. Metabolism primarily occurs in the liver, giving rise to various metabolites, with hydroximetabolite being the most clinically significant due to its strong antimicrobial activity [11,17].

The development of systems for the local modified release of drugs into periodontal pockets is an attractive research area, as it allows reaching higher concentrations of the drug only in the places where it is most needed, minimizing the potential side effects [1,3,7,18]. In this sense, this clinical trial aimed to evaluate (before, during, and after periodontal treatment) the efficacy of semi-solid systems (gels) and films containing metronidazole and metronidazole benzoate as an adjunct to SRP in patients with periodontitis through analysis of the following clinical parameters: gingival crevicular fluid pH, clinical attachment level, probing depth, and bleeding on probing.

## 2. Materials and Methods

### 2.1. Materials

To evaluate the efficacy of systems containing metronidazole and metronidazole benzoate in the treatment of periodontitis, sintering was carried out using Myverol 18-92k^®^ (Kerry, Três Corações, MG, Brazil), lipophilic surfactant (Span 60) HBL 4.7 (Croda Brazil, Campinas, SP, Brazil), metronidazole, metronidazole benzoate (Henrifarma, São Paulo, SP, Brazil), polyethyleneglycol 400 (VETEC Química, Duque de Caxias, Rio de Janeiro), Zein (Sigma-Aldrich, Saint Louis, MO, USA), acetonitrile (JTBaker/Merk, Frankfurt, Germany), and pharmaceutical-grade water: ultrapure MilliQ (Merk, Frankfurt, Germany).

### 2.2. Methods

This randomized, double-blind clinical trial was approved by the Research Ethics Committee of the University of São Paulo at School of Dentistry of Ribeirao Preto (CAAE nº: 2011.1.907.58.0), according to Plataforma Brazil, under the Brazilian Clinical Trials Registry RBR-226F6J and the Universal Trial Number U1111-1246-0723.

#### 2.2.1. Clinical Trial Characteristics

In total, 45 patients were selected from the Ribeirao Preto School of Dentistry–FORP/USP, between 35 and 70 years old, of both genders, based on the following inclusion criteria: patients with stages I or II periodontitis; progression grade A or B; extension and distribution: localized, generalized or incisive molar; probing depth equal to or greater than 5 mm; and not having used antibiotics for at least 3 months. Conversely, the exclusion criteria encompassed severe cardiovascular ailments, pharmacokinetic-related issues such as nephropathies and liver diseases; documented allergic reactions to the components within the tested formulations; systemic conditions potentially affecting disease progression or treatment response (e.g., diabetes and immunological disorders); smoking habits; pregnancy; and lactation.

The blinding was double for the patients and for the statistician, and was maintained due to the fact that the patients were treated without visual contact with the drugs in gel or film form. As the products were coded, the statistician only had access to the raw coded data for each of the 3 groups, and only at the end of the clinical stage.

Following enrollment, 96 teeth were randomly selected and divided (using a computer program, Excel Worksheet—Microsoft Office 2013) into 3 groups according to the local release system (film or gel) associated with the scaling and root planing technique (SRP): GI—SRP, GII—SRP + film, and GIII—SRP + Gel.

#### 2.2.2. Preparation of Film System Containing Metronidazole and Metronidazole Benzoate

The casting/solvent evaporation technique was employed to create a polymeric film consisting of zein, polyethylene glycol 400 (PEG 400) as a plasticizer, and the drugs metronidazole and metronidazole benzoate. To initiate the process, a zein dispersion was prepared by combining zein with a hydroethanolic solution (1:9 ratio) and subjecting it to magnetic stirring at 400 rpm for 3 h. PEG 400 was then introduced into the mixture and stirred for an additional 30 min. Subsequently, the drugs were incorporated into the dispersion and stirred for another 30 min. A total of 80 g of the polymeric dispersion was deposited onto a 12 × 12 cm polystyrene plate, after which it was dried in a climatic chamber at 50 °C and 50% relative humidity for 16 h. The composition of the dispersion for film preparation is detailed in Table 1. Following the drying process, the resulting film contained a drug concentration of 4.5% metronidazole and 9.0% metronidazole benzoate.

#### 2.2.3. Preparation of a Semi-Solid (Gel) System Containing Metronidazole and Metronidazole Benzoate

The preparations involved heating glyceryl monolinoleate (Myverol^®^ 18-92k) along with the surfactant in a thermostatic water bath at a temperature of 40/45 °C. Once heated, manual homogenization was performed using a glass rod, after which the drugs were introduced into the mixture while stirring. The final step involved adding distilled water to the formulation at the same temperature. The resulting mixture was then left undisturbed for 24 h at a controlled temperature of 25 °C ± 1. This period of rest was necessary before any further manipulation. The composition of the semi-solid formulation can be found in Table 2.

#### 2.2.4. Storage Conditions of Film and Gel Formulations

The compositions were stored at ambient temperature for the film and refrigerated for the gel, and were monitored for a minimum duration of six months. Both formulations maintained visual integrity, exhibiting no alterations in color or consistency, and the drug concentrations remained stable. With respect to the film, the complete absence of water contributes significantly to its stability. Furthermore, the active ingredients possess antimicrobial properties, which aid in the prevention of contaminant proliferation.

#### 2.2.5. Introduction of the Polymeric Devices Into the Periodontal Pockets

The polymeric devices were introduced into the periodontal pockets with the aid of clinical tweezers for the film (Figure 1) or by means of a syringe with a blunt needle for the gel (Figure 2).

Pilot studies were conducted to assess both the syringeability of the gel, defined as the ability to extrude the gel from a syringe without fracturing its components, and the suitability of the needle volume for passive introduction into periodontal pockets. Needles with a length of 20 mm were evaluated across various gauges (in mm): 0.6, 0.8, 0.9, and 1.2. It was found that the most appropriate diameter for the application of this semi-solid formulation was 0.9 mm. Needles with diameters larger than 1.2 mm or smaller than 0.6 mm affected the force required for the extrusion process, making them less suitable for this application.

#### 2.2.6. Quantification of Drug Present in the Gingival Crevicular Fluid

The quantification of metronidazole in the base forms (MDZ) and benzoate (BMDZ) were evaluated 48 h after application of the formulations from filter paper strips (Periopaper^®^) adsorbed of crevicular fluid was performed using high-performance liquid chromatography (HPLC). With the inclusion of BMDZ in the formulations, it was necessary to alter the existing method to make simultaneous quantification of the two forms of the asset possible. Thus, the method described by Sato et al. (2008) [20] was adequate for this purpose. For extraction, each strip was conditioned in a 2.0 mL microtube, added with 0.25 mL of methanol and submitted to an ultrasonic bath for 30 min. The paper strip was removed, and the sample was submitted to centrifugation at 15,000× *g* for 15 min and filtered using a 4 mm diameter nylon syringe filter and 0.22 μm pore size.

The analysis was performed on a Shimadzu chromatograph Prominence model with SIL-20AHT automatic sampler, CBM-20A controller, LC-20AT pump system, and SPD-M20 diode arrangement detector. A C18 reverse phase column (Gemini model, Phenomenex) was used, with dimensions of 4.6 mm × 25 cm and a particle diameter of 5 μm, coupled to a pre-column C18 (Gemini 4.0 × 3.0 mm). As the mobile phase, a mixture of methanol and ultrapure water at 40 °C was used, in a linear gradient of 40 to 90% of methanol in 15 min, at a flow rate of 0.8 mL/min, and a temperature of 40 °C. The sample volume injected was 30 μL and detection at 320 nm. The results were obtained through Software LabSolutions (Shimadzu, Kyoto, Japan).

For quantitation of total active drug form (MDZ), BMDZ values were expressed as MDZ. The conversion was performed based on the straight-line equations of the drug analytical curves and the molecular weight ratio between MDZ (MM = 171.15) and BMDZ (MM = 275.26).

#### 2.2.7. In Vivo Evaluation of Periodontal Intrapocket pH

Samples were collected with pH indicator strips and placed in 1.2 mL microtubes with TE buffer solution (10 mM Tris-HCl, 1 mM EDTA, pH 7.6), where pH indicators were inserted.

#### 2.2.8. Evaluation of Clinical Parameters Indicative of Periodontitis

The clinical parameters of probing depth (PD) (from the free gingival margin to the most apical portion of the gingival sulcus), clinical attachment level (CAL) (from the amelocemental junction to the most apical portion of the gingival sulcus or periodontal pocket), and bleeding on probing (BP) (presence or absence of bleeding within 15 s after probing) were evaluated in the periodontal pockets selected for the study using the Florida Probe^®^ periodontal probing system at times: T0 = initial; T7 = after 7 days; T15 = after 15 days; T30 = after 30 days; T60 = after 60 days, and T90 = after 90 days.

#### 2.2.9. Statistical Analysis

The mean data of PD, CAL, BP, and pH for each participant were subjected to statistical analysis using the Wald test within the framework of generalized estimation equation (GEE) to assess differences between treatment groups and time points. Subsequent multiple comparisons were conducted with Bonferroni correction, and statistical significance was established at a 5% significance level.

## 3. Results

This study comprised a total of 45 participants, with each participant receiving a single type of treatment for all affected teeth. The SRP group included 14 teeth from 7 participants. The group SRP + Film treatment involved 41 teeth from 23 participants. The SRP + Gel group consisted of 32 teeth from 15 participants. Therefore, an average of 2 teeth per participant were assessed.

In cases where some participants missed certain consultations, their results were still included in the analysis, except for the specific consultations they were absent from.

### 3.1. Quantification of Drug in Gingival Fluid

The presence of the drug was assessed 48 h after administration and then re-evaluated after 7 days. The drug was not detected at the 7-day mark. For the 48 h assessment the drug concentration found was 49.0 µg after application of the film and 21.77 µg after application of the gel, and it was estimated that the volume of fluid collected by the periopaper was 1 μL (Table 3). There was no significant difference between the means evaluated by the t test (*p* = 0.1306).

### 3.2. Evaluation of Periodontal Intrapocket pH and Clinical Parameters

The analysis of periodontal intrapocket pH revealed no statistical differences between the groups (*p* = 0.569), the evaluated time points (*p* = 0.786), and the interaction between group and time was also not significant (*p* = 0.429) (Figure 3).

The analysis of the clinical attachment level (CAL) results indicated that the Group factor was highly significant (*p* < 0.001), whereas Time (*p* = 0.555) and the Group vs. Time interaction (*p* = 0.136) were not statistically significant. Comparing the different groups, it was observed that the SRP + Gel group exhibited significantly lower values compared to the SRP + Film group (*p* < 0.001) (Figure 4).

Regarding probing depth (PD), Time (*p* < 0.001) and the Group vs. Time interaction (*p* = 0.002) were found to be significant, whereas the Group factor (*p* = 0.504) did not show significant results. Analyzing the Group vs. Time interactions, no significant differences were observed between the groups at each time point. However, within each group, a decrease in PD over time was observed for the SRP group, with T0 displaying higher values compared to T7, T30, T60, and T90 (*p* < 0.001). Additionally, T7 exhibited higher values than T30 (*p* = 0.019), T60 (*p* < 0.001), and T90 (*p* = 0.033). Furthermore, T15 showed higher values than T60 (*p* < 0.001) and T90 (*p* = 0.002). For both the SRP + Film and SRP + Gel groups, higher PS values were observed at T0 (*p* < 0.001) compared to T7, T15, T30, T60, and T90, with no significant differences found among the other time points (Figure 5).

In the case of bleeding on probing (BP), Group (*p* = 0.011), Time (*p* < 0.001), and the Group vs. Time interaction (*p* = 0.027) were all significant. Analyzing the Group vs. Time interaction, no significant differences were found between the groups at any given time. When comparing the time points within each group, the SRP group did not exhibit a significant change in BP over the evaluated times. On the other hand, both the SRP + Film and SRP + Gel groups displayed a significant decrease in BP at all time points compared to T0 (*p* < 0.001). Starting from T7, no significant differences were observed between the time points for both the SRP + Film and SRP + Gel groups (*p* > 0.05) (Figure 6).

## 4. Discussion

Chronic inflammatory periodontal disease is caused by host immune responses to periodontal microorganisms [1,2,3,21,22,23,24,25,26,27]. Mechanical periodontal treatment can often be sufficient to improve or resolve the clinical picture in most cases of periodontal disease. However, in cases of non-response to conventional mechanical therapy, adjunctive antimicrobial agents administered locally or systemically, may increase the effect of therapy in specific situations [1,3,22,23,25,26,28]. In order to allow the elimination or reduction of periodontopathogenic species [1,2,3,21,22,23,24,26], pharmacological agents applied to the periodontal intrapocket should reach the site of action, maintaining adequate concentrations for a sufficient period of time and in some situations remaining after scaling and root planing [1,9,22,24].

In this study, metronidazole associated with metronidazole benzoate with a hydrophobic characteristic was used as the antimicrobial agent. This association allowed the detection and quantification of the drug for longer periods, after 48 h of periodontal treatment, when compared with studies using only metronidazole, maximum of 24 h [9,20,23,24,26]. Another factor that contributed to the maintenance of intrapocket drug was the placement of surgical periodontal cement in both groups after conventional periodontal treatment. The surgical cement acted as a physical barrier, preventing or hindering the exit of the drug shortly after its insertion. This is in contrast with Miani et al., 2012 [9] in which no physical barrier was installed after insertion of the gel containing metronidazole, which may have facilitated the removal of the drug through the salivary fluids, reducing their permanence in the evaluated sites.

With respect to pH, patients with periodontal disease presented an alkaline pH (around 8), which is in agreement with Khan et al., 2017 [24] and Eggert et al., 1991 [29] who state that oral pH varies in different sites and conditions of oral health and may be indicative of disease presence. It is suggested that these results can be explained by the calcification mechanism of the bacterial biofilm, since the increase in the salivary pH favors the precipitation of crystals of calcium and phosphate on the softened surface of the biofilm [24]. In the present study, it was not possible to find a statistically significant difference between the pH value, the group to which the patient belonged (SRP, film + SRP, or SRP + SRP), time (initial or final), and the interaction between group and time. It is worth mentioning that in the present study, only one operator, previously calibrated, in order to avoid errors and interferences in the analysis of the results since the pH indicator tapes have small variations in the color gradation, performed the pH measurement as well as having varied decimal values.

In this study, there was no statistical difference between the groups evaluated, neither in the interaction between group × time. However, in relation to the times, there was a statistical difference; the longer the time elapsed after the treatment, the lower the probing depth indices. Furthermore, it was possible to verify that in the group that received the drug through the film, there was a decrease in the probing depth values, but not in a constant way.

Although the interaction was not significant, a slightly different behavior was observed in each group, in which the SRP + film group presented a progressive reduction in probing depth up to 15 days with maintenance after this period. In the SRP + gel group, the mean reduction in probing depth also occurred in a progressive manner. However, after T60, there was an increase in PD. In relation to the control group (SRP), a difference in probing depth of approximately 1 mm more was observed in the analysis of the mean values, when compared to the groups that received film or gel, 7 days after the treatment.

It is suggested that the release of metronidazole intrapocket in this group was reduced or ceased due to the shelf life of the film [9,24]. The results of this study corroborate the works of Riep; Purucker; Bernimoulin, 1999 [30]; Jansson; Bratthall; Soderholm, 2003 [31]; Leiknes et al., 2007 [32]; Singh et al., 2009 [33]; and Khan et al., 2016 [24] that when evaluating the SRP associated with the local application of metronidazole, there was no statistical difference in the clinical parameters evaluated between the groups.

According to the Periodontology Academy Association (1997) [34], periodontal health can be observed clinically from four to six weeks after treatment, in which the clinical improvement of the parameters evaluated can be verified [1,2,23,24,25,26]. For Griffiths et al., 2000 [35], this period corresponds to the beginning of the maintenance phase (on average, 90 days after treatment), as can also be observed in this study.

It is the parameter that determines the extent of destruction of periodontal tissues and should be performed in at least two different periods [1,2,24,25]. In this study, there was a statistically significant difference for both groups evaluated with a progressive decrease in CAL after 90 days of treatment for the SRP and SRP + Film groups. However, it was not possible to observe a statistical difference between the times and between the groups x time interaction. These results contrast with the works of Miani et al., 2012 [9]; Riep; Purucker; Bernimoulin, 1999 [30]; Jansson; Bratthall; Soderholm, 2003 [31]; Leiknes et al., 2007 [32]; and Singh et al., 2009 [33], that when evaluating the clinical attachment level in patients with periodontal disease, there was no statistical differences between the evaluated groups. On the other hand, the group that received only conventional periodontal treatment presented a lower variation in CAL values over the 90 days, showing greater stability and predictability for this system.

Bleeding on probing, whether spontaneous or provoked, is one of the earliest and most reliable indicators of periodontal inflammation [1,2,23,24,25]. While the absence of bleeding generally signifies healthy periodontal tissue, Salvi et al. (2002) [36] demonstrated that even in an intact periodontium, applying a force greater than 0.25 N during probing can induce bleeding.

In our study, 89.29% of patients exhibited bleeding on probing at baseline (T0), which significantly decreased to 25.71% after 90 days of treatment. These findings are consistent with the work of Armitage, 2000 [37] and 2003 [38] and Khan et al., 2017 [24], who also observed a progressive reduction in bleeding following scaling and root planing (SRP) over a 30-day period.

Our results further highlighted a statistically significant difference in bleeding reduction between the groups treated with SRP combined with a drug-delivery film or gel compared to SRP alone. Specifically, the SRP + Film group showed a marked reduction in bleeding as early as 15 days post-treatment, which was sustained through day 90. The SRP + Gel group exhibited significant reductions starting at day 30, with these effects persisting until day 90. In contrast, the SRP-only group did not show a significant reduction in bleeding.

The association of local drug delivery with conventional periodontal treatment clearly enhanced the therapeutic outcomes, as evidenced by the reduction in bleeding on probing. The faster onset of bleeding reduction in the film group compared to the gel group may be attributable to the more rapid drug release observed in film-based treatments. Our in vivo evaluation of semi-solid systems (gels) and films containing metronidazole salt and metronidazole benzoate conjugate confirmed the presence of the drug within periodontal pockets up to 48 h post-treatment. The pH values post-treatment ranged from 6 to 8, creating a favorable environment for periodontal healing.

Patients who received the SRP + Film treatment also experienced a progressive reduction in probing depth, which was maintained up to 90 days after treatment. Similarly, a significant reduction in bleeding on probing was observed at T15 and T30 in the experimental groups, a trend not seen in the SRP-only group. Regarding clinical attachment levels, the SRP + Gel group achieved the most favorable outcomes, with statistically significant improvements compared to the other groups.

Despite these promising results, several limitations must be considered. The study’s sample size, while sufficient for initial analysis, could be expanded in future research to improve the generalizability of the findings. Additionally, the follow-up period was limited to 90 days; extending this period would allow for a better understanding of the long-term effects of these treatments. Future studies should also explore the underlying mechanisms that differentiate the drug release profiles between gel and film formulations and their impact on long-term periodontal health. Furthermore, investigating the potential synergistic effects of combining these drug delivery systems with other adjunctive therapies could provide valuable insights into optimizing treatment protocols for chronic periodontitis.

## 5. Conclusions

According to the results obtained in this study, overall, it can be concluded that, in relation to clinical parameters, both therapies (gel and film) were effective in promoting the improvement of periodontal health when compared to SRP-alone therapy. Furthermore, it was possible to observe that the association of metronidazole gel therapies with SRP is a promising therapy since it was able to promote reduction in probing depth, gain in the clinical attachment level and lower indices of bleeding on probing, suggesting its use as adjuvant to periodontal therapy.

## Figures and Tables

**Figure 1 pharmaceutics-16-01108-f001:**
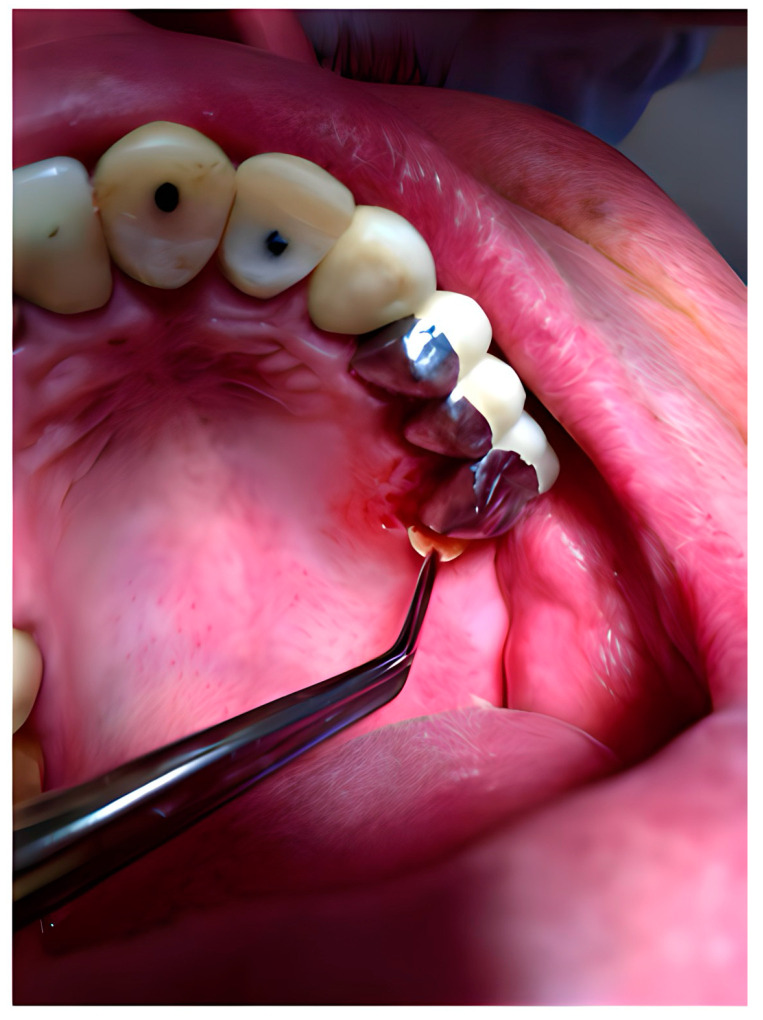
Film in situ application.

**Figure 2 pharmaceutics-16-01108-f002:**
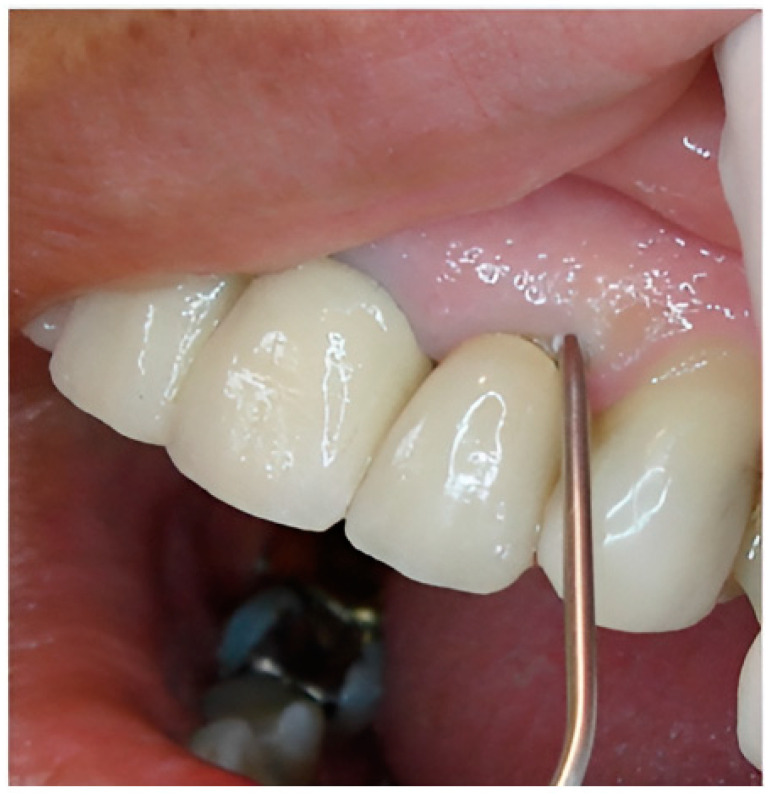
Gel in situ application.

**Figure 3 pharmaceutics-16-01108-f003:**
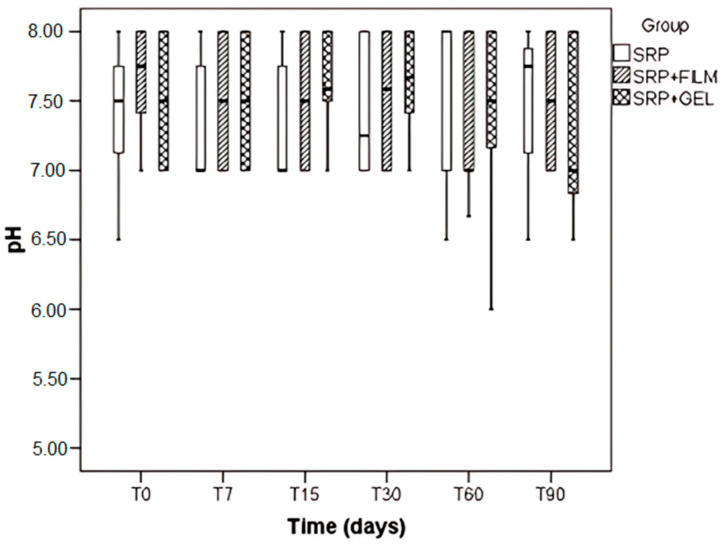
Boxplot of the pH values evaluated in the different types of treatment SRP, SRP + Film, and SRP + Gel according to time in days.

**Figure 4 pharmaceutics-16-01108-f004:**
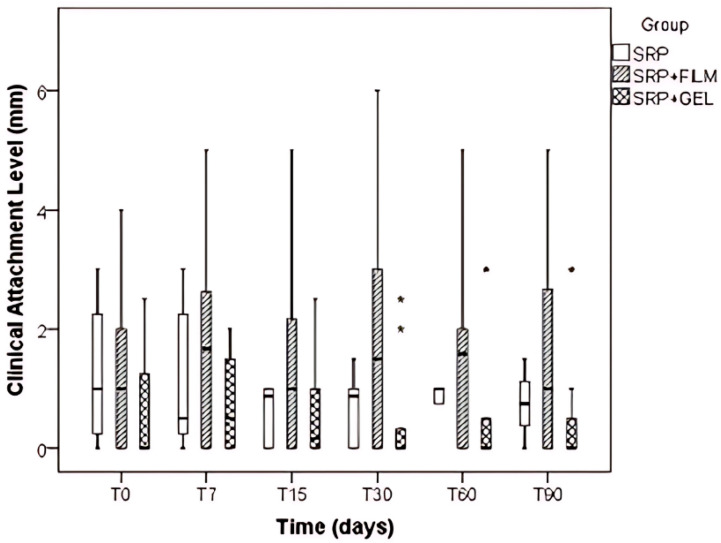
Boxplot of the clinical attachment level (CAL) value evaluated in the different types of treatment SRP, SRP + Film, and SRP + Gel according to time in days. The asterisks (*) indicate data outliers.

**Figure 5 pharmaceutics-16-01108-f005:**
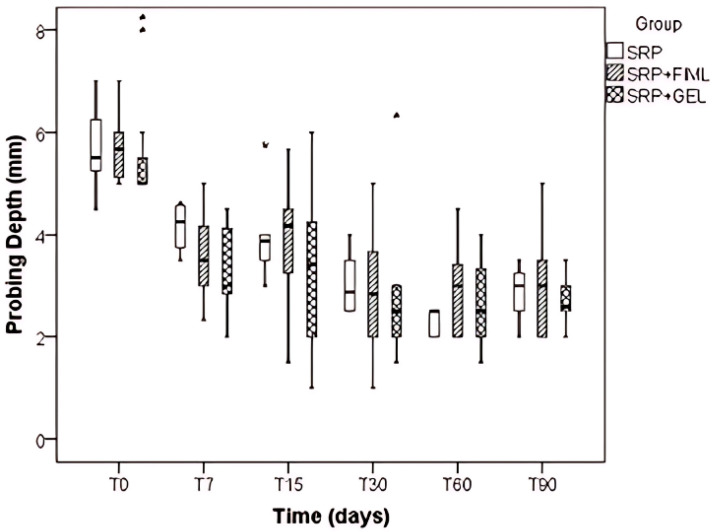
Boxplot of the probing depth (PD) value evaluated in the different types of treatment SRP, SRP + Film, and SRP + Gel according to time in days. The asterisks (*) indicate data outliers.

**Figure 6 pharmaceutics-16-01108-f006:**
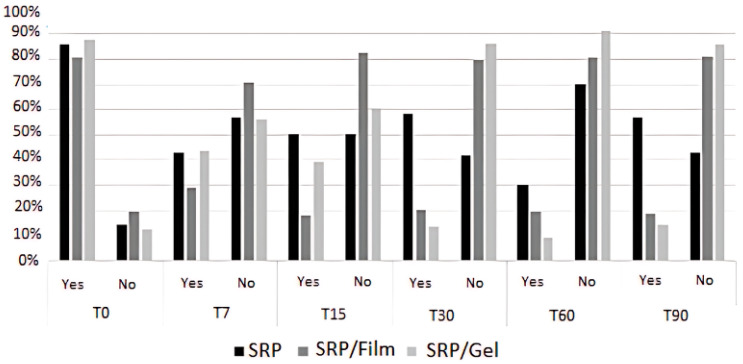
Mean values of the clinical parameter of bleeding on probing (BP), in percentage (%), evaluated in the different types of treatment SRP, SRP + Film, and SRP + Gel according to the time (days).

**Table 1 pharmaceutics-16-01108-t001:** Composition of the formulation for preparing the film system.

Component	Concentration (%)
Zein	6.00
Plasticizer	1.20
Metronidazole	0.37
Metronidazole Benzoate	0.75
Hydroethanolic solution (1:9)	91.7

**Table 2 pharmaceutics-16-01108-t002:** Composition of the formulation for preparing the semi-solid (gel) system.

Component	Concentration (%)
Myverol^®^ 18-92k	56.57
Lipophilic surfactant SPAN 60 (sorbitan monostearate) HLB = 4.7	2.33
Metronidazole	5.00
Metronidazole Benzoate	16.10
Water	20.00

**Table 3 pharmaceutics-16-01108-t003:** Descriptive statistics of drug concentration data in gingival fluid, 48 h after application of the formulation.

Formulation	Mean Concentration (µg/mL) ± SD	Median (Range)
Film	49.03 ± 69.87	21.84 (3.95; 244.69)
Gel	21.77 ± 17.49	14.64 (5.75; 57.43)

## Data Availability

The original contributions presented in the study are included in the article, further inquiries can be directed to the corresponding author.

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
