# Peer review of "Metronidazole Modified-Release Therapy Using Two Different Polymeric Systems Gels or Films: Clinical Study for the Treatment of Periodontitis"

_pharmaceutics, 2024, doi:10.3390/pharmaceutics16091108_

Round 1

Reviewer 1 Report

Comments and Suggestions for Authors

This study examined the efficacy of semi-solid systems (gels) and films with metronidazole (MTZ) salt and benzoate combined with scaling and root-planning (SRP) for patients with periodontitis. Effects were evaluated by longitudinal monitoring of clinical parameters probing depth - PD, clinical attachment level – CAL, and bleeding on probing – BP. The pH of gingival crevicular fluid (GCF) before/after treatments, MTZ concentrations, and drug release were also evaluated. Various serious issues needed to be resolved before further consideration.

1.      Table 1, what is the full name for "qsp"?

2.      Table 2, what is the chemical structure or commercial name for the "Lipophilic surfactant"?

3.      Line 213, except for the "specific" consultations.......... What are the "specific consultations"? 

4.      Line 216. The unit used for drug concentration should be ug/mL instead of %.

5.      Table 4. The title should be revised.

6.      Figure 6. Higher resolution was needed.

7.      Line 378-380. There is no data to support the difference in drug released from the film and gel. The only data shown is in Table 3 for the drug released in 48 hours.

Author Response

Dear Reviewer and Editor,

We would like to sincerely thank the reviewers for his/her detailed comments, which were very useful for improving our manuscript. We have taken all these comments into account in the revised version of the paper. The following are the replies to reviewers’ comments. In the answers, the reviewer's comments are in black, and our answers are in bold. We emphasize that we made all the corrections in the text highlighted in yellow.

Best regards.

1. Table 1, what is the full name for "qsp"?

Dear Reviewer, thank you for your consideration. The abbreviation QSP stands for "Quantity Sufficient to." However, we have updated it to 91.7%, which represents the percentage of ethanol used in the study. The change has been made in the text.

2. Table 2, what is the chemical structure or commercial name for the "Lipophilic surfactant"?

Dear Reviewer, thank you for the suggestions provided for our article. The surfactant used was SPAN 60 (Sorbitan monostearate) with an HLB of 4.7.

3. Line 213, except for the "specific" consultations.......... What are the "specific consultations"? 

Dear Reviewer, thank you for your contribution to our article. When a patient missed an appointment, for example, during the second collection, he/she was not excluded from the sample. The statistical model used allowed for the entry of "not assessed" on the day of the missed appointment, and the data obtained from subsequent sections were tabulated and used in the analysis.

4. Line 216. The unit used for drug concentration should be ug/mL instead of %.

Dear Reviewer, thank you for all your contributions to this article.
Sorry for the typographical error. It has been corrected in the text to µg/mL.

5. Table 4. The title should be revised.

Dear Reviewer, we have made changes to the title of the manuscript to better understand the study.

6. Figure 6. Higher resolution was needed.

Dear Reviewer, according to your suggestion, the quality of figures were solved

7. Line 378-380. There is no data to support the difference in drug released from the film and gel. The only data shown is in Table 3 for the drug released in 48 hours.

Dear Reviewer, thank you for all your corrections and work in reviewing our article. In the study, both the 48-hour and 7-day time points were evaluated. Since no drug was detected at the 7-day mark, quantification was only possible at the 48-hour time point and was thus included in the work.

Reviewer 2 Report

Comments and Suggestions for Authors

This manuscript investigates the use of metronidazole modified-release formulations for periodontal treatment, offering valuable insights into the potential benefits of localized drug delivery systems in clinical practice. The study is well-designed, with a solid methodology and a clear presentation of results. However, to strengthen its scientific rigor and enhance its contribution to the field, the manuscript needs to address several key issues: it should clarify the rationale behind specific experimental choices, expand the discussion on the clinical significance of the findings, and improve the clarity and quality of all figures. These improvements are necessary for the manuscript to be accepted for publication.

1. The manuscript does not discuss the potential impact of storage conditions on the stability of film and gel formulations.

2. The use of a double-blind design is laudable, but the manuscript should detail how the blinding was maintained throughout the study.

3. The manuscript mentions the use of a blunt needle to apply the gel, but there is no discussion of whether different needle sizes or application techniques were tested.

4. The manuscript should include a discussion of the potential impact of differences in drug absorption and metabolism among patients on the study results.

5. No other biomarkers of periodontal health were measured (besides clinical attachment levels)

6. No detailed analysis of the physical properties of the films and gels (e.g. adhesion, flexibility) was included.

7. All images in the manuscript are not clear and need to be improved before they can be accepted for publication, including improving image resolution and ensuring labels and legends are clear and easy to read

Comments on the Quality of English Language

Minor editing of English language required

Author Response

Dear Reviewer and Editor,

We would like to sincerely thank the reviewers for his/her detailed comments, which were very useful for improving our manuscript. We have taken all these comments into account in the revised version of the paper. The following are the replies to reviewers’ comments. In the answers, the reviewer's comments are in black, and our answers are in bold. We emphasize that we made all the corrections in the text highlighted in yellow.

Best regards.

1. The manuscript does not discuss the potential impact of storage conditions on the stability of film and gel formulations.

Dear Reviewer, thank you for all your considerations on our article. The compositions were stored at room temperature for the film and in the refrigerator for the gel and were monitored for at least 6 months. Both remained visually intact, with no changes in color or consistency, and the drugs were quantified and remained stable. Regarding the film, the total absence of water contributes to its stability. Additionally, the active ingredients are antimicrobial and help prevent the development of contaminants. We added a subsection to the article to explain the storage conditions.

2. The use of a double-blind design is laudable, but the manuscript should detail how the blinding was maintained throughout the study.

Dear Reviewer, thank you very much for the note. The blinding was double for: i. the patients and for ii. the statistician and was maintained due to the fact that the patients were treated without visual contact with the drugs in gel or film form. As the products were coded, the statistician only had access to the raw coded data for each of the 3 groups, and only at the end of the clinical stage. We have added the information in the methodology.

3. The manuscript mentions the use of a blunt needle to apply the gel, but there is no discussion of whether different needle sizes or application techniques were tested.

Dear Reviewer, thank you very much for the note. We inform that pilot studies were carried out both to evaluate syringeability (ability to extrude the gel from a syringe with a needle, without fractionating the gel components), and to evaluate whether the volume of the needle could be introduced passively into the periodontal pockets. Needles with a length of 20 mm were tested with the following gauges (in mm): 0.6; 0.8; 0.9 and 1.2 and the most suitable diameter was 0.9 mm, for the application of this type of semi-solid formulation, since needles with diameters above (1.2mm) or below (0.6mm) that chosen influenced, the force required to start work.

4. The manuscript should include a discussion of the potential impact of differences in drug absorption and metabolism among patients on the study results.

Dear Reviewer, we sincerely appreciate your continued feedback aimed at improving our manuscript. However, we would like to clarify that the study's objective was not to examine the drug's mechanism of action, given that it is already well-established and widely used. Instead, our focus was on evaluating its effectiveness in treatment by closely monitoring patient outcomes.

5. No other biomarkers of periodontal health were measured (besides clinical attachment levels)

Dear Reviewer, thank you very much for the note. We sought to evaluate as many parameters as possible in the design and development of the study, but we agree that our study has some limitations, including the fact that we did not include other biomarkers of periodontal health, beyond the clinical attachment levels, probing depth and bleeding on probing, as we had the fulcrum of continuing the initial studies of the group with a comparison effect (Sato et al., 2008 and Miani et al., 2012).

6. No detailed analysis of the physical properties of the films and gels (e.g. adhesion, flexibility) was included.

Dear Reviewer, thank you for all your feedback. The compositions were characterized regarding their physical properties. It was observed that the flexibility of the film was adequate for the intended application. Additionally, the gel showed greater adhesiveness in the presence of the drugs compared to the inert (placebo) formulation. This suggests that the presence of the drugs positively influences the gel's adhesion.

7. All images in the manuscript are not clear and need to be improved before they can be accepted for publication, including improving image resolution and ensuring labels and legends are clear and easy to read

Dear Reviewer, according to your suggestion, the quality of figures were solved

Reviewer 3 Report

Comments and Suggestions for Authors

Review 1 pharmaceutics-3158658

The study entitled “Metronidazole modified-release therapy using bioadhesive polymeric systems: pH and bone levels outcomes” by Bastos et al. is an interesting scientific work to improve the evaluation of the clinical efficacy of the administration of two different systems; gels and films, containing both a combination of metronidazole (MTZ) and metronidazole benzoate for the treatment of periodontitis.

This study evaluates the efficacy of semi-solid systems (gels) and films containing a combination of metronidazole (MTZ) and metronidazole benzoate after scaling and root planning (SRP) for periodontitis in 45 patients. For its evaluation, a clinical trial is carried out comparing the efficacy of SRP alone, SRP + films and SRP + gels. Different parameters such as pH and metronidazole release are evaluated at 48 h after the administration of each treatment. In addition, different clinical parameters; probing depth (PD), clinical attachment level (CAL) and bleeding on probing (BP) are evaluated throughout the treatment.

Title

1-The title does not reflect the study carried out. You must write a new title that includes the pharmaceutical forms studied and the performance of a clinical study.

A possible example:

Metronidazole modified-release therapy using two different polymeric systems gels or films: Clinical study for the treatment of periodontitis.

You must carefully select the title of this work.

Abstract

2-Pg. 1. Line 17-18

The paragraph

“This study evaluated the efficacy of semi-solid systems (gels) and films with metronidazole (MTZ) salt and benzoate combined with scaling and root-planing (SRP) for periodontitis.”

It should be replaced by

“This study evaluated the efficacy of semisolid systems (gels) and films containing a combination of metronidazole (MTZ) and metronidazole benzoate after scaling and root-planing (SRP) for periodontitis.”

2. Materials and Methods

3-In this section you must include the following subsections:

2.1. Material

In this section you must include the supplier (city and country) of all raw materials: Metronidazole, Metronidazole benzoate, zein, polyethyleneglycol 400, Myverol® 18-92K, indicate the specific lipophilic surfactant used including characteristics (HLB), HPLC grade Acetonitrile, pharmaceutical grade water…

2.2. Methods

4-Pg.3 Line 125 you must include a subsection

2.2.1. Clinical Trial characteristics

Including from Pg.2 line 96 to Pg. 3 Line 112

5-Section Pg. 3 Line 113

2.1. Preparation of film system containing metronidazole and metronidazole benzoate

It should be:

2.2.2. Preparation of film system containing metronidazole and metronidazole benzoate

6-Pg. 3 Line 118. You indicate

“a zein dispersion was prepared by combining zein with a hydroethanolic solution (9:1 ratio)”. Wouldn't it be solution (1:9 ratio)?

However, Pg. 3 Line 135 in table 1 indicates that 90% ethanol was used

Please confirm and leave the correct one.

7-Pg. 3 Line 138. You indicate

2.2.3. Preparation of semi-solid (gel) system containing metronidazole and metronidazole benzoate

8-Pg. 4 Line 144. Table 2. Indicates Lipophilic surfactant you must replace it with the name of the surfactant used (indicating its HLB).

9-Pg. 4 Line 151. You indicate

2.2.4. Quantification of drug present in the gingival crevicular fluid

-In this section Figures 1 and 2 must be deleted.

10-Pg. 5 Line 182 you have to include the calibration line (R2 value) and detection limits (LD) and quantification (LQ).

11-Pg. 5 Line 185. You indicate

2.2.5. In vivo evaluation of periodontal intrapocket pH

12-Pg.6 Line 190

2.2.6. Evaluation of clinical parameters indicative of periodontitis

Results

13-Pg. 7 Line 2297

Figure 3 doesn't look good (blurry). You have to do it again, improving its resolution.

14-Pg. 8. Line 233

Delete Table 4. The results of Table 4 are already included in figures 4, 5 and 6. In addition, the columns have no space and the description of this table is not correct.

15-Pg.9 Linea 240; Pg. 10 Line 255 and Pg. 10 Line 284

Figures 4, 5 and 6 doesn't look good (blurry). You have to do it again, improving its resolution.

Discusion

16-I would delete the paragraph from Pg. 12. Line 378-381 (However, when the release form of the ...) and include the following paragraph below

Pg. 12 Line 374

Patients who received gel and film showed a reversion in the situations where bleeding on probing was present. In the case of the film, a significant reduction in bleeding after probing could be observed soon after T15, while in the gel it occurred after T30. These efficacy results could be consistent with the drug delivery results which showed faster drug delivery in film treatments compared to gel treatment.

Author Response

Dear Reviewer and Editor,

We would like to sincerely thank the reviewers for his/her detailed comments, which were very useful for improving our manuscript. We have taken all these comments into account in the revised version of the paper. The following are the replies to reviewers’ comments. In the answers, the reviewer's comments are in black, and our answers are in bold. We emphasize that we made all the corrections in the text highlighted in yellow.

Best regards.

Title

1. The manuscript does not discuss the potential impact of storage conditions on the stability of film and gel formulations. The title does not reflect the study carried out. You must write a new title that includes the pharmaceutical forms studied and the performance of a clinical study.

A possible example:

Metronidazole modified-release therapy using two different polymeric systems gels or films: Clinical study for the treatment of periodontitis.

You must carefully select the title of this work.

Dear Reviewer, we appreciate your suggestion to improve the title of our manuscript. Based on your suggestions, we have made the necessary changes.

Abstract

2. Pg. 1. Line 17-18

The paragraph

“This study evaluated the efficacy of semi-solid systems (gels) and films with metronidazole (MTZ) salt and benzoate combined with scaling and root-planing (SRP) for periodontitis.”

It should be replaced by

“This study evaluated the efficacy of semisolid systems (gels) and films containing a combination of metronidazole (MTZ) and metronidazole benzoate after scaling and root-planing (SRP) for periodontitis.”

Dear Reviewer, we appreciate your suggestion to improve the abstract of our manuscript. Based on your suggestions, we have made the necessary changes.

Materials and Methods

3. In this section you must include the following subsections:

2.1. Material

In this section you must include the supplier (city and country) of all raw materials: Metronidazole, Metronidazole benzoate, zein, polyethyleneglycol 400, Myverol® 18-92K, indicate the specific lipophilic surfactant used including characteristics (HLB), HPLC grade Acetonitrile, pharmaceutical grade water…

Dear Reviewer, we have implemented the suggested changes to enhance the clarity and organization of the methodological sections.

2.2. Methods

4. Pg.3 Line 125 you must include a subsection

2.2.1. Clinical Trial characteristics

Including from Pg.2 line 96 to Pg. 3 Line 112

Dear Reviewer, we have implemented the suggested changes to enhance the clarity and organization of the methodological sections.

5. Section Pg. 3 Line 113

2.1. Preparation of film system containing metronidazole and metronidazole benzoate

It should be: 

2.2.2. Preparation of film system containing metronidazole and metronidazole benzoate

Dear Reviewer, we have implemented the suggested changes to enhance the clarity and organization of the methodological sections.

6. Pg. 3 Line 118. You indicate

“a zein dispersion was prepared by combining zein with a hydroethanolic solution (9:1 ratio)”. Wouldn't it be solution (1:9 ratio)?

However, Pg. 3 Line 135 in table 1 indicates that 90% ethanol was used

Please confirm and leave the correct one.

Dear Reviewer, thank you once again for all the corrections and suggestions. I apologize for the typographical error; the correct ratio is indeed 1:9. We have made the changes in the article.

7. Pg. 3 Line 138. You indicate

2.2.3. Preparation of semi-solid (gel) system containing metronidazole and metronidazole benzoate

Dear Reviewer, we have implemented the suggested changes to enhance the clarity and organization of the methodological sections.

8. Pg. 4 Line 144. Table 2. Indicates Lipophilic surfactant you must replace it with the name of the surfactant used (indicating its HLB).

Dear Reviewer, thank you for the suggestions provided for our article. The surfactant used was SPAN 60 (Sorbitan monostearate) with an HLB of 4.7.

9. Pg. 4 Line 151. You indicate

2.2.4. Quantification of drug present in the gingival crevicular fluid

-In this section Figures 1 and 2 must be deleted.

Dear Reviewer, we have implemented the suggested changes to enhance the clarity and organization of the methodological sections.

10. Pg. 5 Line 182 you have to include the calibration line (R2 value) and detection limits (LD) and quantification (LQ).

Dear Reviewer, thank you again for the suggestions provided for our article. For quantitation of total active drug form (MDZ) BMDZ values were expressed as MDZ. The conversion was performed based on the straight -line equations of the drug analytical curves and the molecular weight ratio between MDZ (MM=171.15) and BMDZ (MM=275.26). The regression line equation for MDZ was y=94010x-48337, range from 0.5 to 80ug/mL, R2=0.99 and for BMDZ it was y=86941x+5985.2, R2=0.99 range from 0.805 to 128.8ug/mL.

11. Pg. 5 Line 185. You indicate

2.2.5. In vivo evaluation of periodontal intrapocket pH

Dear Reviewer, we have implemented the suggested changes to enhance the clarity and organization of the methodological sections.

12. Pg.6 Line 190

2.2.6. Evaluation of clinical parameters indicative of periodontitis

Dear Reviewer, we have implemented the suggested changes to enhance the clarity and organization of the methodological sections.

Results

13. Pg. 7 Line 2297 

Figure 3 doesn't look good (blurry). You have to do it again, improving its resolution.

Dear Reviewer, according to your suggestion, the quality of figures were solved

14. Pg. 8. Line 233

Delete Table 4. The results of Table 4 are already included in figures 4, 5 and 6. In addition, the columns have no space and the description of this table is not correct.

Dear Reviewer, we sincerely appreciate your continued suggestions to enhance our manuscript. We have implemented the suggested changes and deleted table 4.

15. Pg.9 Linea 240; Pg. 10 Line 255 and Pg. 10 Line 284

Figures 4, 5 and 6 don't look good (blurry). You have to do it again, improving its resolution.

Dear Reviewer, according to your suggestion, the quality of figures were solved

Discussion

16. I would delete the paragraph from Pg. 12. Line 378-381 (However, when the release form of the ...) and include the following paragraph below

Pg. 12 Line 374

Patients who received gel and film showed a reversion in the situations where bleeding on probing was present. In the case of the film, a significant reduction in bleeding after probing could be observed soon after T15, while in the gel it occurred after T30. These efficacy results could be consistent with the drug delivery results which showed faster drug delivery in film treatments compared to gel treatment.

Dear Reviewer, we sincerely appreciate your continued suggestions to enhance our manuscript. In accordance with your recommendation, we have made the necessary revisions to the discussion section.

Reviewer 4 Report

Comments and Suggestions for Authors

This study evaluated the efficacy of semi-solid systems (gels) and films with metronidazole (MTZ) salt and benzoate combined with scaling and root-planing (SRP) for periodontitis. And 45 patients with stage I or II periodontitis were enrolled. Overall, this study is interesting and well performed with a series of characterizations. However, there still some issues need to be addressed.

1.      The logic of the introduction and discussion section is poor, especially the discussion section. There is too many information without well-organized. For example, the bleeding data was repeated discussed in Line 364-377. The whole discussion section is highly recommended to be re-organized.

2.      Which kind of surfactant was used in the gel formulation.

3.      How to ensure the prepared gel and film preparations are sterile?

4.      ANOVA is always used for the statistical analysis, why Wald test was used in this study.

5.      The annotate of Table 4 was not consistent with the content.

6.      The quality of Figure 4-6 is poor.

Author Response

Dear Reviewer and Editor,

We would like to sincerely thank the reviewers for his/her detailed comments, which were very useful for improving our manuscript. We have taken all these comments into account in the revised version of the paper. The following are the replies to reviewers’ comments. In the answers, the reviewer's comments are in black, and our answers are in bold. We emphasize that we made all the corrections in the text highlighted in yellow.

Best regards.

1. The logic of the introduction and discussion section is poor, especially the discussion section. There is too many information without well-organized. For example, the bleeding data was repeated discussed in Line 364-377. The whole discussion section is highly recommended to be re-organized. 

Dear Reviewer, thank you for your valuable feedback and suggestions for improving our manuscript. In response, we have revised the discussion section to enhance clarity and comprehension.

2. Which kind of surfactant was used in the gel formulation.

Dear Reviewer, thank you again for all the suggestions made. The type of surfactant used in the gel formulation was SPAN 60 (Sorbitan monostearate). We made the change in the article to make it clearer to understand.

3. How to ensure the prepared gel and film preparations are sterile?

Dear Reviewer, in the specific case of our study we did not sterilize the formulations because we did not see the need, considering that the active ingredient is an antibiotic, the components (therapeutic adjuvants) of both the film and the gel had a chemical degree of purity and were manipulated in a formulation laboratory with strict controls of the best pharmaceutical handling practices, and considering that the application site (periodontal pockets) is not sterile. We also followed the same principles as the group's previous studies (Sato et al., 2008 and Miani et al., 2012).

4. ANOVA is always used for the statistical analysis, why Wald test was used in this study.

Dear Reviewer, thank you for all the corrections made. The reason we used the Wald method for analysis was due to the absence of a normal distribution of the data. The Wald test is the non-parametric equivalent of ANOVA.

5. The annotate of Table 4 was not consistent with the content.

Dear Reviewer, we sincerely appreciate your continued suggestions to enhance our manuscript. We have implemented the suggested changes and deleted table 4, because all data was already on figures 4, 5 and 6.

6. The quality of Figure 4-6 is poor.

Dear Reviewer, according to your suggestion, the quality of figures were solved

Round 2

Reviewer 1 Report

Comments and Suggestions for Authors

The authors have properly addressed the issues raised.

Reviewer 2 Report

Comments and Suggestions for Authors

The author has addressed all my concerns, and it can be published now.

Comments on the Quality of English Language

N/A

Reviewer 4 Report

Comments and Suggestions for Authors

The authors have revised the manuscript according to the reviewer's comments, and the quality of this manuscript has been improved, it should be published now.